# Heterologous Expression and Biochemical Characterization of a New α-Amylase from *Nocardiopsis aegyptia* HDN19-252 of Antarctic Animal Origin

**DOI:** 10.3390/md23040159

**Published:** 2025-04-04

**Authors:** Fuhao Liu, Xiangnan Zheng, Wenhui Liao, Xingtao Ren, Chuanteng Ma, Guojian Zhang, Qian Che, Tianjiao Zhu, Wenxue Wang, Tao Zhang, Feng Han, Dehai Li

**Affiliations:** 1Key Laboratory of Marine Drugs, Ministry of Education, School of Medicine and Pharmacy, Ocean University of China, 5 Yushan Road, Qingdao 266003, China; 17863905950@163.com (F.L.); zhengxiangnan@qidu-pharma.com (X.Z.); liaowenhui726@163.com (W.L.); xingtao.ren@gmail.com (X.R.); ma_chuanteng@163.com (C.M.); zhangguojian@ouc.edu.cn (G.Z.); cheqian064@ouc.edu.cn (Q.C.); zhutj@ouc.edu.cn (T.Z.); bx_wwx@163.com (W.W.); 2Laboratory for Marine Drugs and Bioproducts, Qingdao Marine Science and Technology Center, Qingdao 266237, China; 3Shandong Qidu Pharmaceutical Co., Ltd., 17 Hongda Road, Linzi District, Zibo 255400, China

**Keywords:** α-amylases, *Nocardiopsis aegyptia*, halotolerance

## Abstract

α-Amylases, catalyzing starch degradation, serve as vital biocatalysts in industrial and pharmaceutical applications. This study identified a new α-amylase, Alphaz, from *Nocardiopsis aegyptia* HDN19-252 of Antarctic animal origin, achieving heterologous expression in *Escherichia coli*. Phylogenetic analysis confirmed its classification into the GH13_5 subfamily of glycoside hydrolases. Recombinant Alphaz exhibited optimal activity at 40 °C/pH 8.0 while maintaining stability across 0–30 °C and pH 6.6–9.6. Its distinctive halotolerant properties included full activity retention in 0.6 M NaCl and >60% efficiency in salt-free conditions. The enzyme exhibits tolerance to K^+^, Ca^2+^, and Fe³^+^ while demonstrating specific inhibition by Cu^2+^/Zn^2+^. With its heterologously validated functional properties, Alphaz emerges as a programmable enzymatic tool offering advantages in sustained-release formulation quality control, targeted prodrug modification, and precision medicine applications, thereby enabling sustainable biomanufacturing solutions that harmonize process reliability with environmental compatibility.

## 1. Introduction

α-Amylases (EC 3.2.1) are glycoside hydrolases that catalyze the cleavage of α-1,4-glycosidic bonds in starch and related polysaccharides. Within the Carbohydrate-Active Enzymes (CAZy) database, these enzymes are classified under diverse families and subfamilies. The majority belong to the GH13 family, which includes subfamilies such as GH13_1 (classical α-amylases), GH13_5 (bacterial α-amylases), and GH13_24 (bacterial maltogenic α-amylases) [1]. Notably, certain extremophilic α-amylases are categorized into the GH57 family [2]. Further classification of microbial α-amylases is based on their optimal operating conditions, including acidophilic, alkaliphilic, thermophilic, and psychrophilic variants [3]. This diversity in classification underpins their broad functional adaptability and industrial utility.

α-Amylases are indispensable in multiple sectors, including food processing, detergent formulation, biofuel production, textiles, and sustainable manufacturing [4,5,6,7,8,9,10,11,12]. Their eco-friendly nature aligns with circular economy principles, making them pivotal for green industrial processes. In biopharmaceuticals, α-amylases hydrolyze starch-based fermentation media to enhance yields of antibiotics (e.g., penicillin and streptomycin), vaccines, and therapeutic proteins [13,14]. Additionally, they enable the synthesis of prebiotic oligosaccharides from starch hydrolysates, which are incorporated into nutraceuticals and functional foods to promote gut microbiota balance and immune health [15,16]. Recent innovations in α-amylase applications have significantly expanded their biotechnological and medical potential. The immobilization of α-amylases on solid supports enhances their stability and recyclability, enabling their efficient use in continuous large-scale pharmaceutical production processes [17,18]. Concurrently, starch hydrolysates generated by α-amylase treatment serve as precursors for bioactive peptides with therapeutic properties, including antioxidant, antihypertensive, and immunomodulatory activities, which are increasingly utilized in precision medicine formulations [19]. In drug delivery, α-amylase-responsive systems have been engineered for controlled release of active pharmaceutical ingredients under specific physiological conditions, such as elevated glucose levels in diabetic patients, offering novel strategies for insulin administration and targeted cancer therapies [20,21]. Furthermore, α-amylase-based enzymatic formulations are revolutionizing wound care by degrading polysaccharide-based medical dressings, thereby accelerating tissue regeneration and minimizing infection risks in chronic wound treatment [22,23].

As early as the 1990s, the cold-adapted amylase AMY_PSEHA from the Antarctic bacterium *Alteromonas haloplanctis* was identified. This amylase is famous for its highly efficient catalytic ability at low temperatures [24]. Its low ion requirement can reduce the dependence on cofactors in industry [25]. The low-temperature reaction conditions decrease energy consumption and are suitable for green production processes. It shows unique advantages in the fields of food processing, pharmaceuticals, and environmental protection industries. Coincidentally, we also identified an α-amylase gene from the Antarctic-derived actinomycete *Nocardiopsis aegyptia* HDN19-252. This strain had previously been studied for its natural product potential [26] and was cultured under static rice culture conditions for 30 days, producing a compound with a monosaccharide side chain, whose structure is similar to that of Pluraflavins A [27]. Subsequently, we heterologously expressed the Alphaz gene and purified the resulting protein. After a series of biochemical characterizations, we found that this amylase has a similar low-temperature catalytic ability to AMY_PSEHA. Our discovery has expanded the number of Antarctic-derived amylases.

## 2. Results and Discussion

### 2.1. Strain Identification

The 16S RNA of strain HDN19-252 was sequenced and submitted to the 16S-based EzBioCloud’s identification service. After sequence alignment using MAFFT and subsequent phylogenetic tree construction in MEGA 7.0, it was found that strain HDN19-252 clusters with *Nocardiopsis aegyptia* strain TRM86122 (Accession Number: OK299036.1) on the same evolutionary branch. This finding confirms the species identification of the strain as *Nocardiopsis aegyptia* (Appendix A).

### 2.2. Identification and Sequence Analysis of Alpha-Amylase

After performing similarity analysis by comparing the sequences of multiple amylases with the genome of the strain, the sequence of this amylase (*Alphaz*) was determined (the nucleotide sequence and amino acid sequence of Alphaz are provided in the Appendix A) (Figure 1 and Appendix A). The strain has certain associations with the classic Antarctic bacterial amylase P29957 (AMY_PSEHA). This gene comprises 1824 base pairs, translating into a 607-residue polypeptide containing an N-terminal putative secretion signal peptide (Met1-Thr27) that facilitates extracellular localization. Bioinformatic analyses predicted a molecular mass of 64.0 kDa and an isoelectric point (pI) of 4.19 for the mature enzyme.

### 2.3. Recombinant Expression and Purification of Alphaz

The *Alphaz* (GenBank number: PV223442 ) coding sequence was subcloned into the pET-28a(+) expression vector after truncating the signal peptide coding region (Met1-Thr27) and flanking the gene with hexahistidine ((His)6) affinity tags at both termini. Soluble heterologous expression of recombinant Alphaz was achieved using the pET-28a(+)/*E. coli* BL21(DE3) system. Subsequent purification via immobilized metal affinity chromatography (IMAC) under native conditions efficiently isolated the enzyme from clarified lysate supernatants. Electrophoretic analysis under denaturing conditions (SDS-PAGE) confirmed a final purity >90%, with the purified protein migrating at ~64.6 kDa, consistent with its theoretical molecular mass after signal peptide removal (Figure 2). Enzymatic assays using soluble starch as the substrate demonstrated a specific activity of 1129.41 U/mg. The overall recovery rate reached 23.74%, yielding approximately 9.17 mg of active Alphaz per liter of bacterial culture (Table 1).

### 2.4. Effects of Temperature and pH on Alphaz

According to the results of an optimal temperature test, the optimal reaction temperature for Alphaz is 40 °C. It still retains over 50% enzyme activity at 20 °C. However, the activity of Alphaz drops rapidly when the temperature is above 60 °C, and it essentially loses all enzyme activity at 80 °C. The temperature stability test measured the residual enzyme activity after incubation for 1 h at temperatures ranging from 0 to 70 °C. Alphaz is relatively stable at temperatures below 20 °C, retaining over 90% of its initial activity after 1 h of incubation. It can still maintain more than 80% enzyme activity when the temperature is below 35 °C, and the enzyme is almost inactivated at 40 °C (Figure 3).

Considering that this is a cold-adapted α-amylase, we measured its Tm using the intrinsic fluorescence method, and the Tm was found to be 45.53 °C (Figure 4).

The optimal pH value for the degradation activity of Alphaz against soluble starch was pH 8.0 in 50 mM Na_2_HPO_4_-NaH_2_PO_4_ buffer (Figure 5a). Moreover, Alphaz could keep stable in different buffers of pH 6.6~9.6 (Figure 5b). Alphaz can be stable in neutral and weakly alkaline environments and can be applied in some suitable processes.

### 2.5. Effects of Metal Ions, Chelators, and Surfactants on Alphaz

As shown in Figure 6a, the tested metal ions did not significantly enhance the degradation activity of Alphaz on soluble starch. Mg^2+^, Cu^2+^, EDTA, and SDS can extremely significantly inhibit the activity of Alphaz, while K^+^, Li^+^, Ca^2+^, Ba^2+^, and Fe^3+^ have no significant effect on the activity of Alphaz. The enzyme is relatively sensitive to Cu^2+^, with significant inhibition at 1 mM, and Zn^2+^ can partially inhibit the enzyme activity. In addition, EDTA and SDS can also significantly inhibit the activity of amylase (Figure 6a). The hydrophobic residues of the protein may serve as the basis for withstanding partial SDS [28]. Previous studies have reported that NaCl can enhance the thermal stability of enzymes [29,30]. Therefore, we also measured the effect of NaCl on the thermal stability of the enzyme. However, no enhancement of enzyme stability by NaCl was observed at 37 °C and 40 °C (Figure 6b). The enzyme activity of Alphaz was measured in a 0–1 M NaCl system. The optimal NaCl concentration for Alphaz is 0.6 M (Figure 6c). In the absence of NaCl, the enzyme activity is still above 60% of the relative activity.

This α-amylase exhibits optimal activity at 40 °C (pH 8.0) and has good stability over the range of 0–30 °C and pH 7.6–9.6, making it particularly suitable for low-temperature bioprocessing and precision drug development.

### 2.6. Substrate Specificity of Alphaz

The substrate specificity of Alphaz was evaluated against a panel of polysaccharides, including soluble starch, α-cyclodextrin, wheat flour, potato starch, cornstarch, and pullulan. Enzymatic assays revealed that Alphaz catalyzed the hydrolysis of soluble starch, α-cyclodextrin, wheat flour, potato starch, and cornstarch but showed no detectable activity toward pullulan. Quantitative analysis indicated that soluble starch served as the optimal substrate, with Alphaz displaying a specific hydrolysis rate 1.1-fold higher than that observed for wheat flour, potato starch, and cornstarch, which exhibited comparable degradation efficiencies (~90% of the maximal activity) (Figure 7).

### 2.7. Enzymatic Kinetic Parameters

The enzymatic kinetic parameters were measured and compared with those of the classic cold-adapted amylase from Antarctic bacteria (AMY_PSEHA, Genbank No. P29957) and porcine pancreatic α-amylase [24,25] (Table 2).

Our comparison results indicate that Alphaz has higher catalytic efficiency and substrate binding capacity than the cold-adapted amylase AMY_PSEHA from Antarctic sources and the amylase from porcine sources. Meanwhile, the higher *K*_cat_ value of cold-adapted enzymes is also reflected in our comparison results. We found that for both Alphaz and AMY_PSEHA, their optimal temperatures are lower than the *T*_m_. The potential reasons for this may be that the active site region has higher flexibility, causing the active site to unfold and expose first. Another possible reason is the reduction in the number of salt bridges and hydrophobic interactions. For example, the Antarctic enzyme AMY_PSEHA has fewer salt bridges and hydrophobic clusters than the porcine enzyme, resulting in a lack of rigidity around the active site and its preferential unfolding [25]. The preferential unfolding of the active site of the Antarctic α-amylase is a structural trade-off for its cold adaptability, rooted in the decoupling of high flexibility of the catalytic center from overall stability. This mechanism provides a key target for the rational design of psychrophilic enzymes [31].

### 2.8. The Structure and Catalytic Site Prediction of Alphaz

Sequence comparison and structural analysis revealed that Alphaz possesses the typical catalytic triad of α-amylase, Asp210-Glu234-Asp301 (Figure 8a), as well as the binding sites Tyr254 and His303 (Figure 8b). This is a characteristic feature of the glycoside hydrolase family 13 [24,25]. The proposed mechanism is as follows. In the first step of glycosidic bond hydrolysis, Asp210 acts as the acid catalyst, protonating the oxygen atom (O-) of the substrate glycosidic bond through its proton (H^+^) donor function, thereby weakening the C-O bond. Glu234 activates a water molecule as a nucleophile by deprotonating it to generate a highly nucleophilic hydroxyl ion. Asp301 stabilizes the oxonium ion transition state through hydrogen bonding and electrostatic interactions. The hydroxyl group of Tyr254 forms a hydrogen bond with the substrate, guiding it to correctly orient into the catalytic pocket. The imidazole group of His303 stabilizes the substrate through hydrogen bonding or electrostatic interactions.

When comparing the structure of the Antarctic-derived cold-adapted α-amylase AMY_PSEHA with our obtained α-amylase Alphaz, the root-mean-square deviation (RMSD) of 2.74 Å falls within the range of moderate similarity. Upon comparison, it was found that the N-terminal and C-terminal regions of the two proteins have relatively low similarity, but the overall structure and the core region are highly similar. This indicates that the core folding structures of the two proteins (the arrangement of α-helices and β-sheets) are conserved, but there are local conformational differences.

### 2.9. The Productivity Curve of Alphaz

The enzyme productivity curve shows that during the 0–30 min period, the substrate concentration is much higher than *K*_m_, resulting in a rapid increase in reaction rate. In the 30–120 min stage, the reaction rate gradually decreases, possibly due to the gradual deactivation of the enzyme and the inhibitory effect of the products. From 120 to 240 min, the reaction rate gradually approaches 0, indicating that the enzyme is almost completely deactivated. From 240 to 400 min, the product concentration no longer increases, indicating that the enzyme has lost its activity (Figure 9).

The enzyme demonstrates efficient catalysis in the initial stage (0–30 min), but the reaction cannot proceed to complete substrate conversion due to rapid deactivation and potential product inhibition. Thermal instability is the main bottleneck restricting its industrial application, which needs to be addressed through engineering modification or process optimization.

### 2.10. Discussion

This study systematically analyzed the enzymatic properties of a novel amylase and found that it exhibited good stability within the temperature range of 0–30 °C and the pH range of 6.6–9.6. Notably, the enzyme’s activity was significantly inhibited by Mg^2+^, Cu^2+^, EDTA, and SDS, but it demonstrated strong tolerance to K^+^, Li^+^, Ca^2+^, Ba^2+^, and Fe³^+^. We think that the enzyme may have certain application value.

Firstly, in the application of detergents, traditional detergents often contain metal chelators (such as EDTA) or surfactants (such as SDS), to which this enzyme is sensitive. Therefore, it needs to be combined with the development of environmentally friendly detergents, using EDTA/SDS-free formulations (for example, systems based on citrates or biosurfactants) [31]. This direction is in line with the current market trend for green detergents. Secondly, it can also be applied in specific scenarios in food processing, such as the alkaline extraction of corn starch (pH 8–9). However, its insufficient thermal stability (inactivation above 37 °C) limits its application in high-temperature sterilization or cooking processes.

Although the enzyme shows certain potential, its insufficient thermal stability (as the productivity curve also indicates) and sensitivity to Mg^2+^/Cu^2+^ remain the main bottlenecks for industrial application. In the future, attempts can be made in enzyme molecular modification, development of EDTA/SDS-free industrial formulations, and enhancement of enzyme operational stability through immobilization techniques.

## 3. Materials and Methods

### 3.1. Reagents

Phanta max super-fidelity DNA polymerase (P505-d1) was purchased from Vazyme (Nanjing, China). Soluble starch was purchased from Solarbio (Beijing, China). Wheat flour, potato starch, and cornstarch were all purchased locally. Pullulan were purchased from Opal (Zhengzhou China). TIANamp bacteria DNA kit (Tiangen, Beijing, China) was used to extract the bacterial genome. HisTrap HP column and Superdex Peptide 10/300 GL were purchased from GE Healthcare (Pittsburgh, PA, USA).

### 3.2. Isolation of Nocardiopsis Aegyptia HDN19-252

Ophiuroid samples were collected at 61°42′28′′ S, 57°38′22′′ W in Antarctica. The broken animal ophiuroid samples were cultured on ISP2 solid medium by the dilution spread plate method. After 7 days of cultivation, single colonies were picked and streaked in three zones to obtain pure single colonies (Appendix A).

### 3.3. Identification of Strain HDN19-252

The isolated strain HDN19-252 was identified using 16S rDNA sequence using EzBioCloud’s identification service (https://www.ezbiocloud.net, accessed on 28 September 2019). Multiple sequence alignment was performed using MAFFT version 7 (https://mafft.cbrc.jp/alignment/server/, accessed on 26 March 2025), and the phylogenetic tree was constructed using MEGA 7.0.

### 3.4. Sequence Analysis of Alphaz

The genomic DNA of *Nocardiopsis aegyptia* HDN19-252 was isolated using the Tianamp Bacterial DNA Extraction Kit (Tiangen Biotech, China), followed by whole-genome sequencing on the Illumina NovaSeq platform (Novogene Bioinformatics Technology Co., Ltd., Beijing, China). The genome of the *Alphaz* was obtained from the Antarctic-derived *Nocardiopsis aegyptia* HDN19-252. We applied for a GenBank number in the GenBank database of the National Center for Biotechnology Information (NCBI) in the United States. Phylogenetic relationships were reconstructed using the neighbor-joining method in MEGA 7.1. For bioinformatic characterization of Alphaz, the molecular weight (Mw) and isoelectric point (pI) were predicted via the ExPASy Compute Mw/pI tool (https://web.expasy.org/compute_pi/, accessed on 12 September 2024). Signal peptide identification and cleavage site prediction were conducted using SignalP 5.0 (http://www.cbs.dtu.dk/services/SignalP/, accessed on 12 September 2024), and conserved domains were mapped via the NCBI Conserved Domain Database (CDD) (https://www.ncbi.nlm.nih.gov/Structure/cdd/wrpsb.cgi, accessed on 12 September 2024). Multiple sequence alignment with GH_13 family enzymes was visualized using ESPript 3.0 (https://espript.ibcp.fr/ESPript/ESPript/, accessed on 15 November 2024) to highlight conserved structural motifs.

### 3.5. Recombinant Expression and Purification of Alphaz

The Alphaz gene sequence (excluding the signal peptide encoding region) was amplified via PCR using primers 252.1-F (5′-TGGTGGTGGTGGTGctcgagGTTCCGCCAGACCGGGG-3′) and 252.1-R (5′-TGCCGCGCGGCAGCcatatgACCCCCGCGGCGCCCGTC-3′), engineered with *XhoI* and *NdeI* restriction sites (lowercase bold), respectively. PCR amplification was performed with genomic DNA from *Nocardiopsis aegyptia* HDN19-252 as the template. The resulting amplicon was digested with *NdeI* and *XhoI* (Thermo Fisher Scientific, Waltham, USA), gel-purified, and directionally cloned into the linearized pET-28a(+) vector (Novagen, Beijing, China) to generate the recombinant plasmid pET-28a-Alphaz. This construct was transformed into *Escherichia coli* BL21(DE3) competent cells (Vazyme Biotech, Nanjing, China) to establish the expression strain.

For recombinant protein production, the strain was cultured in Luria–Bertani (LB) medium (50 μg/mL kanamycin) at 37 °C with agitation (200 rpm) until reaching mid-log phase (OD_600_ = 0.4–0.6). Alphaz expression was induced with 0.1 mM isopropyl-β-D-thiogalactopyranoside (IPTG; Sangon Biotech, Shanghai, China), followed by incubation at 25 °C for 24 h. Cells were harvested by centrifugation (8000× *g*, 30 min, 4 °C) and resuspended in ice-cold binding buffer (20 mM PB, 500 mM NaCl, pH 7.0) at a 1:10 (v/v) ratio. Cell lysis was achieved using a high-pressure homogenizer (JNBIO, Guangzhou, China) at 4 °C, and the soluble fraction (crude extract) was obtained by centrifugation (12,000× *g*, 30 min, 4 °C).

The crude extract was loaded onto a pre-equilibrated Ni-Sepharose column (Cytiva, Marlborough, USA) for immobilized metal affinity chromatography (IMAC). Bound proteins were eluted with a linear imidazole gradient (20–200 mM) in binding buffer. Purified Alphaz was analyzed by SDS-PAGE (12% gel) with Coomassie Brilliant Blue R-250 staining (Bio-Rad, Hercules, USA) to confirm molecular weight (~64 kDa) and purity (>90%). Protein concentration was quantified using a BCA assay kit (EpiZyme, Cambridge, USA) following the manufacturer’s protocol.

### 3.6. Enzyme Activity Determination of Alphaz

The α-Amylase activity was quantified using a modified 3,5-dinitrosalicylic acid (DNS) method (Appendix B). Briefly, 490 μL of substrate solution (0.2% w/v soluble starch in 50 mM phosphate buffer, pH 6.8) was pre-warmed at 37 °C for 5 min, followed by addition of 10 μL purified enzyme solution to initiate the reaction. After 5 min of incubation at 37 °C, the enzymatic hydrolysis was terminated by rapid addition of 750 μL DNS reagent (Sigma-Aldrich, St. Louis, USA) and subsequent boiling for 5 min [31]. The reaction mixture was cooled in an ice bath to room temperature, and the absorbance of the reducing sugars (maltose equivalents) was measured at 540 nm using a spectrophotometer (Thermo Scientific, USA). A blank control, prepared by replacing the enzyme with an equal volume of buffer, was processed identically. One unit (U) of α-amylase activity was defined as the amount of enzyme required to liberate 1 μmol of reducing sugars per minute under the assay conditions (37 °C, pH 6.8).

### 3.7. Biochemical Characterization of Alphaz

To determine the temperature optimum, recombinant Alphaz (10 μL) was incubated with 490 μL of 0.2% (w/v) soluble starch substrate (50 mM Na_2_HPO_4_-NaH_2_PO_4_ buffer, 100 mM NaCl, pH 7.0) at 0–70 °C (10 °C increments) for 5 min. Reactions were quenched with 750 μL of 3,5-dinitrosalicylic acid (DNS; Sigma-Aldrich, USA), boiled for 5 min, and cooled to ambient temperature. Absorbance at 540 nm was measured using a spectrophotometer (Thermo Scientific NanoDrop 2000). Heat-inactivated enzyme (10 min boiling) served as the negative control [32,33].

The *T*_m_ was measured using a differential scanning calorimeter (Malvern PEAQ-DSC, Worcestershire, UK). Prior to scanning, all solutions were degassed by stirring under vacuum for 15 min. The scanning was conducted by increasing the temperature from 25 °C to 100 °C at a rate of 1 °C min^−1^, and the final results were plotted using GraphPad Prism 10.0 [31].

To determine the optimal pH, buffered substrates (0.2% soluble starch) were prepared in Na_2_HPO_4_-Citrate (pH 3.0–8.0), Na_2_HPO_4_-NaH_2_PO_4_ (pH 6.0–8.0), Tris-HCl (pH 7.6–10.0), and Glycine-NaOH (pH 8.6–10.6). Enzyme reactions (10 μL enzyme + 490 μL substrate) were conducted at the optimal temperature, terminated with DNS reagent, and quantified as above. Relative activity was normalized to the maximum absorbance (100%) under optimal conditions.

To assess pH stability, enzyme aliquots were pre-incubated with buffers (490 μL) spanning pH 3.0–10.6 at 0 °C for 12 h. Residual activity was measured by adding 100 μL of pre-treated enzyme to 900 μL of 0.2% soluble starch substrate (optimal pH buffer) and incubating at the optimal temperature for 5 min. Reactions were processed identically to prior assays, with heat-inactivated enzyme as the control. Enzyme activity (U/mL) was calculated based on maltose standard curves, and relative activity was expressed as a percentage of the untreated control (100%).

To evaluate thermal stability, Alphaz enzyme solutions were exposed to temperatures ranging from 0 to 70 °C (10 °C increments) for 1 h in a circulating water bath (JULABO, Allentown, USA), followed by rapid cooling in an ice-water bath for 10 min. Residual activity was assayed by mixing 10 μL of heat-treated enzyme with 490 μL of 0.2% (w/v) soluble starch substrate (prepared in optimal pH buffer containing 100 mM NaCl) and incubating at the optimal temperature (40 °C) for 5 min. Reactions were quenched with 750 μL of 3,5-dinitrosalicylic acid (DNS; Sigma-Aldrich, St. Louis, USA), boiled for 5 min, and absorbance quantified at 540 nm using a spectrophotometer (Thermo Scientific NanoDrop 2000). Heat-inactivated enzyme (10 min boiling) served as the negative control. Relative activity was expressed as a percentage of the untreated enzyme (100%).

For modulator studies, soluble starch substrates (0.2% w/v) were supplemented with 1 mM metal ions, 1 mM EDTA, or 0.1% SDS (w/v) in optimal pH buffer. Enzyme solutions (10 μL) were reacted with 490 μL of each substrate at 40 °C for 5 min. Reactions were processed identically to thermal stability assays. Activity in the presence of modulators was normalized to the untreated control (100% activity) [34].

### 3.8. Substrate Specificity Analysis of Alphaz

Substrate specificity assays were performed by preparing 0.2% (w/v) solutions of α-cyclodextrin, wheat starch, potato starch, and pullulan in a buffer adjusted to the enzyme’s optimal pH and NaCl concentration. Each substrate solution (490 μL) was mixed with 10 μL of diluted enzyme and incubated at the optimal temperature for 5 min. Reactions were terminated by adding 750 μL of 3,5-dinitrosalicylic acid (DNS) reagent (Sigma-Aldrich, St. Louis, USA), followed by boiling for 5 min. After cooling to ambient temperature, the absorbance of reducing sugars was measured at 560 nm using a spectrophotometer (Thermo Scientific NanoDrop 2000). Heat-inactivated enzyme (boiled for 5 min prior to reaction) served as the negative control. Relative enzyme activity was normalized to 100% based on the maximal absorbance observed under optimal conditions [35].

### 3.9. Bioinformatic Analysis

Protein sequence was submitted to the AlphaFold 3 [36] online server (https://deepmind.google/technologies/alphafold/alphafold-server/, accessed on 16 March 2025) for three-dimensional structural prediction. PyMOL 3.1.4 software was used to visualize the structure of the protein, and to analyze the surface charge and hydrophilic distribution of the protein. The amino acid sequence of Alphaz, along with α-amylase from other sources, was uploaded to EMBL-EBI (https://www.ebi.ac.uk/services/ accessed on 18 March 2025) for multiple sequence alignment [37].

### 3.10. The Determination of the Productivity Curve of Alphaz

Prepare the substrate solution by dissolving 10 g of soluble starch in 500 mL of preheated pH 8.0 Tris-HCl buffer solution (60 °C). After stirring to dissolve, cool the solution to 37 °C and adjust the volume to 500 mL to obtain a final starch concentration of 20 mg/mL. The enzyme solution is prepared by diluting the original solution. Mix 1 mL of the original solution with an appropriate amount of 1 mM NaCl buffer solution to obtain a working solution of 0.01 mg/mL. The reaction system is initiated by mixing 45 mL of substrate solution pre-warmed to 37 °C with 5 mL of enzyme solution (total volume 50 mL) and is continuously reacted in a magnetic stirrer (300 rpm) at 37 °C. Samples of 1 mL are taken at preset time points (0, 5, 10, 20, 30, 60, 120, 180, 240, 300, 360 min), and immediately 1 mL of pre-cooled anhydrous ethanol is added to terminate the reaction. Take 0.1 mL of the supernatant after centrifuging the terminated sample (8000 rpm, 5 min), add 0.3 mL of DNS reagent, and color-develop in a boiling water bath for 5 min. After cooling, the absorbance at 540 nm is measured to calculate the amount generated [38].

## 4. Conclusions

A new α-amylase, designated Alphaz, was characterized from the marine-derived actinomycete *Nocardiopsis aegyptia* HDN19-252. This enzyme demonstrates biotechnologically relevant properties, including an optimal activity at 40 °C and pH 8.0, with cold adaptability (retaining >80% activity at 30 °C) and pH stability across 6.6–9.6. Alphaz exhibits a certain degree of halotolerance, maintaining activity even in the presence of 2.0 M NaCl. It displays functional resilience to pharmaceutically prevalent ions (K^+^, Li^+^, Ca^2+^, Ba^2^⁺, Fe³⁺), minimizing interference from excipients. Conversely, sensitivity to Cu^2^⁺, Zn^2^⁺, EDTA, and SDS (0.1% w/v) facilitates activity modulation during production workflows. These attributes position Alphaz as a robust biocatalyst for drug synthesis, particularly in salt-enhanced formulations (e.g., parenteral solutions) or processes involving metal-sensitive APIs. Its alkaline pH stability enables applications in enteric-coated formulations targeting intestinal delivery, while low-temperature operability (≤30 °C) preserves thermolabile substrates. The enzyme’s compatibility with industrial processing parameters underscores its potential for scalable biomanufacturing of precision therapeutics.

## Figures and Tables

**Figure 1 marinedrugs-23-00159-f001:**
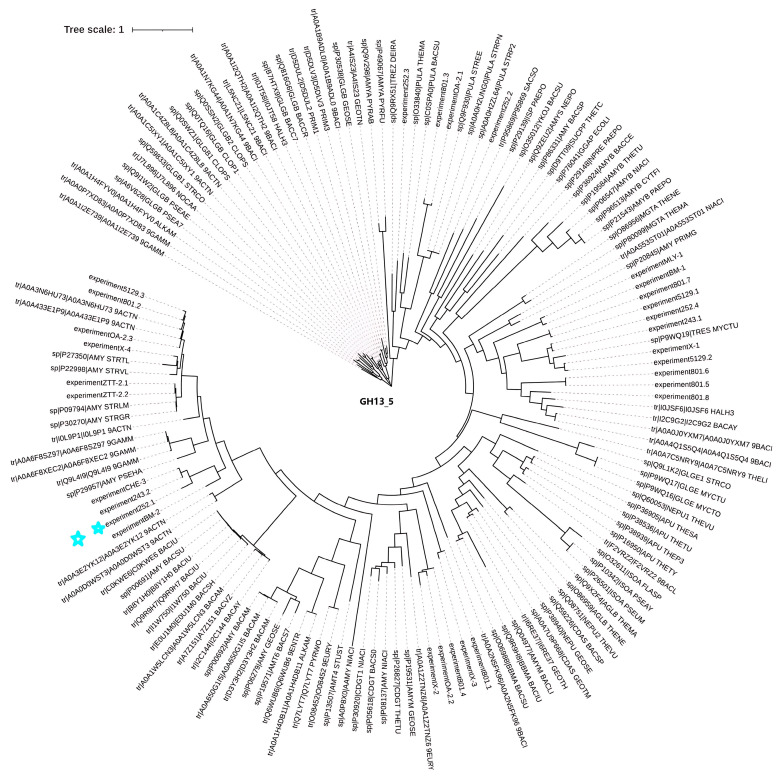
Phylogenetic tree of Alphaz and other members of GH13_5. Amino acid sequences were used for this analysis. The distances on the branches indicate the reliability of the corresponding branches. The closer the distance, the more reliable it is. Alphaz is marked with a blue star.

**Figure 2 marinedrugs-23-00159-f002:**
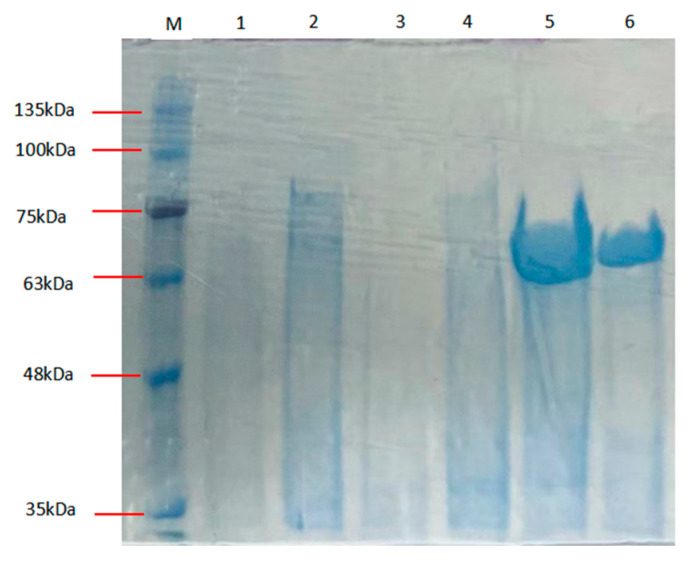
SDS-PAGE of Alphaz. Lane M, protein marker; lanes 5, 6, purified Alphaz.

**Figure 3 marinedrugs-23-00159-f003:**
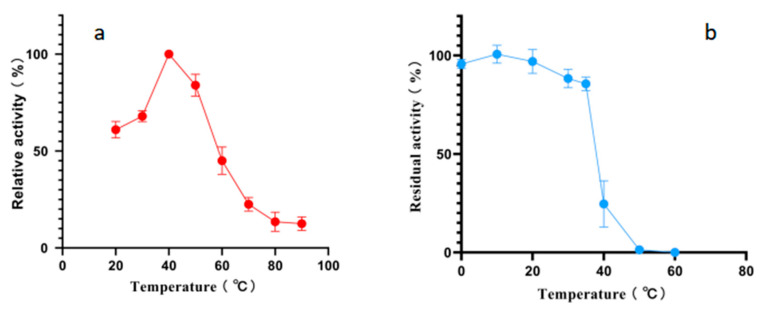
Effects of temperature on Alphaz. (**a**) Optimal temperature of Alphaz. (**b**) The thermostability of Alphaz. Incubated for 5 min; the DNS method was used for detection; error bars indicate standard deviation (*n* = 3).

**Figure 4 marinedrugs-23-00159-f004:**
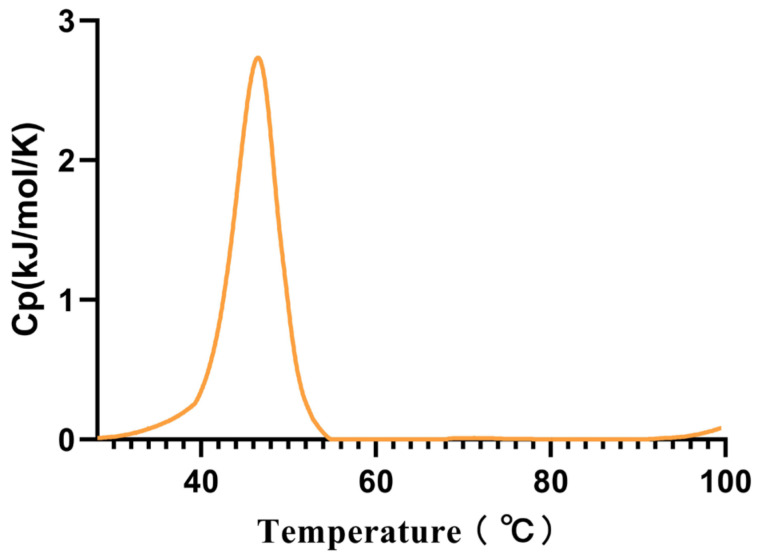
The melting temperature (*T*_m_) of Alphaz. Heated from 25 °C to 100 °C within 75 min. The graph was generated using differential scanning calorimetry (DSC).

**Figure 5 marinedrugs-23-00159-f005:**
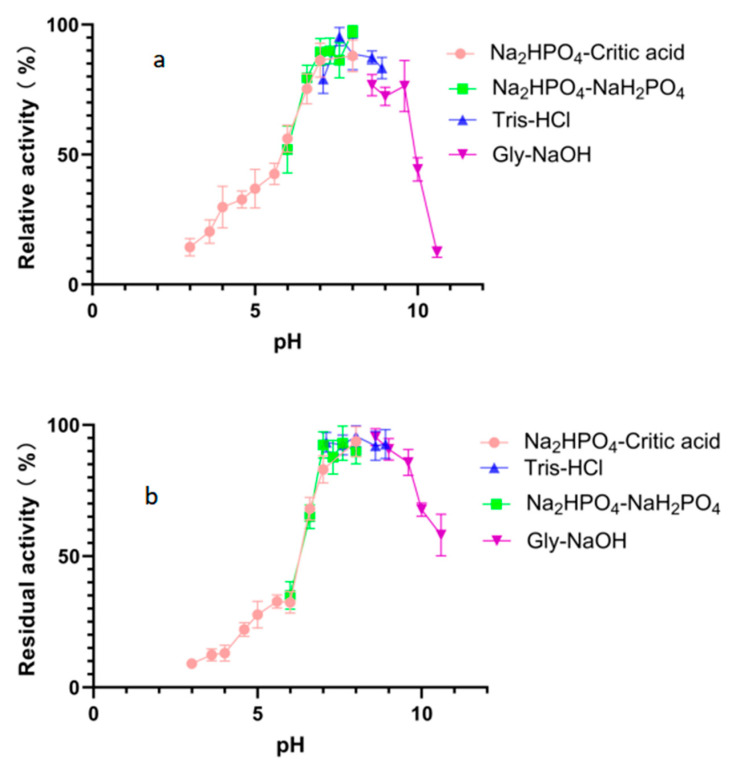
Effects of pH on Alphaz. (**a**) Optimal pH of Alphaz. (**b**) pH stability of Alphaz. Soluble starch was used as the substrate. The activity of Alphaz at the optimal pH and temperature was defined as 100%. Incubated at 40 °C for 5 min; the DNS method was used for detection; error bars indicate standard deviation (*n* = 3).

**Figure 6 marinedrugs-23-00159-f006:**
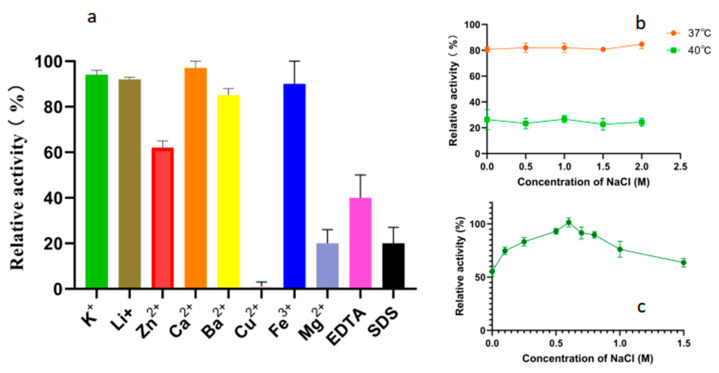
Effects of metal ions, chelators, and detergents on the activity of Alphaz. (**a**) Effects of metal ions, chelator (1 mM), and surfactant (SDS, 0.1%, *w*/*v*). (**b**) Effects of NaCl concentrations (0~1 M). (**c**) The impact of NaCl on enzyme stability. Incubated at 40 °C, pH = 8.0 for 5 min; the DNS method was used for detection; error bars indicate standard deviation (*n* = 3).

**Figure 7 marinedrugs-23-00159-f007:**
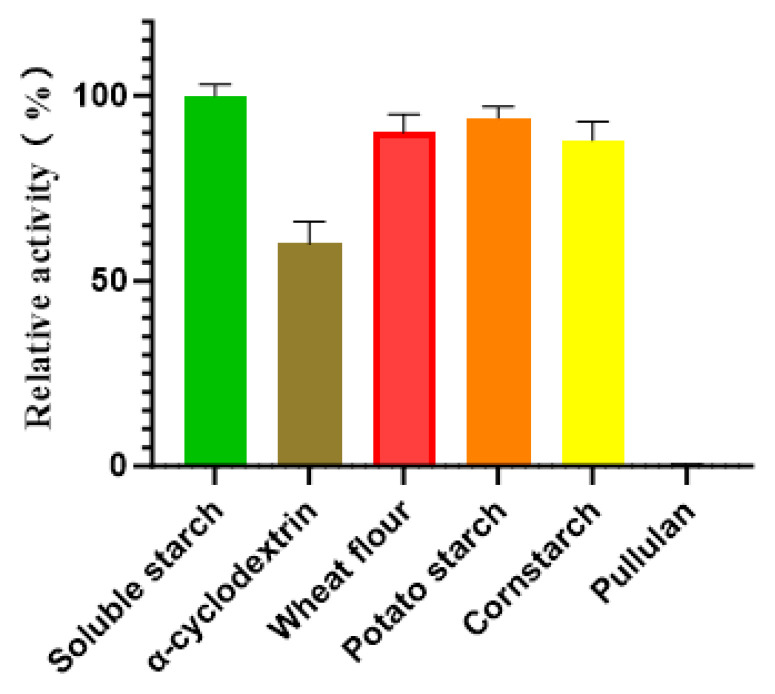
Substrate specificity of Alphaz. The activity of Alphaz towards soluble starch was defined as 100%. Incubated at 40 °C, pH = 8.0 for 5 min; the DNS method was used for detection; error bars indicate standard deviation (*n* = 3).

**Figure 8 marinedrugs-23-00159-f008:**
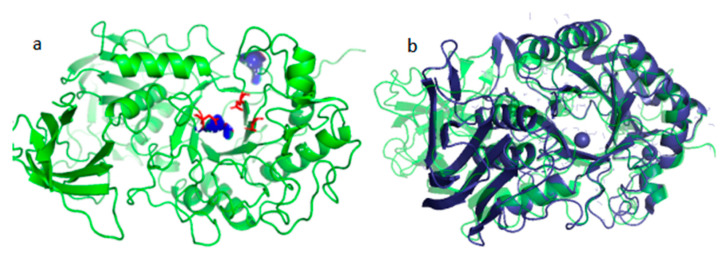
Structural prediction of Alphaz and its catalytic binding sites (**a**), structural comparison of Alphaz (green) with AMY_PSEHA (dark blue) (**b**).

**Figure 9 marinedrugs-23-00159-f009:**
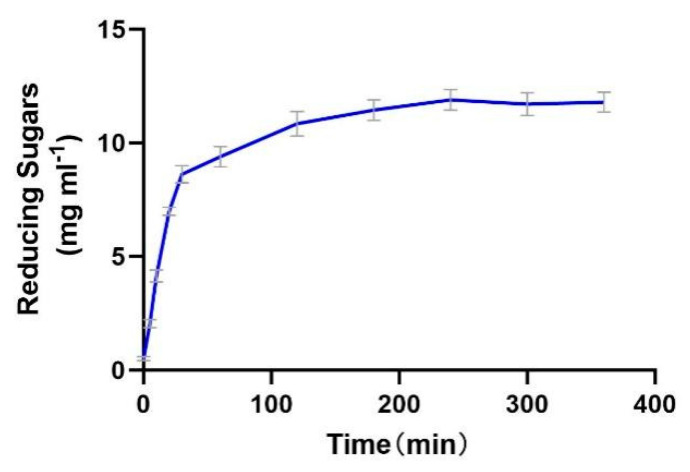
The productivity curve of Alphaz. Incubated for 400 min; the DNS method was used for detection; error bars indicate standard deviation (*n* = 3).

**Table 1 marinedrugs-23-00159-t001:** Summary of the purified Alphaz (*n* = 3).

Step	Specific Activity (U/mg)	Total Protein (mg)	Fold Purification	Yield (%)
Fermentation broth	286.25 ± 25.74	152.40 ± 13.72	1	100
Nickel column	1129.41 ± 109.83	9.17 ± 0.86	3.95 ± 0.38	23.74 ± 2.26

The DNS method was used for this detection. Enzyme activity was measured using 0.2% soluble starch as substrate (20 mM PB, pH 7.0) under optimal conditions. The volume of the initial fermentation broth was 400 mL, and the volume of the concentrated crude extract was 40 mL.

**Table 2 marinedrugs-23-00159-t002:** Comparison of functional characteristics and parameters of enzymes.

Enzyme	*K*_m_ (mg/mL)	*k*_cat_ (s^−^¹)	*k*_cat_/*K*_m_ (s^−1^ · mg^−1^ · mL)	*T_m_* (°C)	*T_opt_* (°C)
Porcine	1.12	326	292	-	50
*A. haloplanctis* AMY_PSEHA	1.27	1363	1075	40	30
Alphaz	1.01	1216.67	1204.62	45.53	40

## Data Availability

The data presented in this study are available in this article and Appendix A.

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
