# Peer review of "Heterologous Expression and Biochemical Characterization of a New α-Amylase from Nocardiopsis aegyptia HDN19-252 of Antarctic Animal Origin"

_marinedrugs, 2025, doi:10.3390/md23040159_

Round 1

Reviewer 1 Report

Comments and Suggestions for Authors

The article "Heterologous expression and biochemical characterization of a new α-amylase from Nocardiopsis aegyptia HDN19-252 of Antarctic animal origin" is devoted to a relevant topic. The scientific novelty of the article is beyond doubt. The manuscript is written quite well, the text is well structured.

I have only minor comments:

  1. In order to increase the interest of readers and the scientific significance of the article, the authors should compare their results with the data of other authors regarding α-amylases from other sources – purity, substrate specificity, temperature and pH optima, thermal stability, pH stability, effects of metal ions, chelators, and surfactants. Ideally, this should be done in the form of a table, but it is also possible in the text of the manuscript.
  2. Based on the comparison results (see point 1), the advantages of α-amylase from Nocardiopsis aegyptia HDN19-252 which is expressed to Escherichia coli should be highlighted compared to other α-amylases. In addition, it would be good to write a few lines about the cost-effectiveness of this amylase.

Technical shortcomings:

  1. Word hyphenation should be avoided in the article title.
  2. All Latin names of microorganisms should be italicized, in particular Escherichia coli (3rd line of Abstract), Nocardiopsis aegyptia (2nd line of Conclusion).
  3. The captions to Figure 1 should be moved to the same page where the Figure itself is located.

Author Response

Dear Reviewer,

Thank you sincerely for taking the time to review our manuscript and for providing such thoughtful and detailed feedback. Your expertise and insights have been instrumental in refining our work, and we are grateful for the opportunity to improve the paper based on your suggestions.

Response to your comments:

Comment 1:In order to increase the interest of readers and the scientific significance of the article, the authors should compare their results with the data of other authors regarding α-amylases from other sources – purity, substrate specificity, temperature and pH optima, thermal stability, pH stability, effects of metal ions, chelators, and surfactants. Ideally, this should be done in the form of a table, but it is also possible in the text of the manuscript.

Response1: The comparison of the enzyme's properties with α-amylases from other sources has been placed in Section 2.7 of the main text, where we have made a preliminary comparison with α-amylases from mammalian and Antarctic sources. Thank you very much for your valuable suggestions.

Comment 2:Based on the comparison results (see point 1), the advantages of α-amylase from Nocardiopsis aegyptia HDN19-252 which is expressed to Escherichia coli should be highlighted compared to other α-amylases. In addition, it would be good to write a few lines about the cost-effectiveness of this amylase.

Response2: First of all, thank you very much for your suggestions. Following your advice, we compared the enzyme kinetic constants of our amylase with those from other sources and found that it shows a certain degree of advantage in catalytic efficiency. This part of the content is also in Section 2.7.

In addition, we have also revised the formatting errors you mentioned.

Once again, we deeply value the time and effort you dedicated to reviewing our work. Your constructive critique has not only improved this manuscript but also provided us with valuable guidance for future research. Please do not hesitate to contact us if additional revisions or explanations are needed.

Sincerely,
Dehai Li
On behalf of all authors

Reviewer 2 Report

Comments and Suggestions for Authors

The manuscript presented here aims to describe the properties of a new type of alpha-amylase produced in the E. coli expression system. Since alpha-amylases have a wide range of practical applications, the development of a new tool is always a welcome development.

The main expectation from this work is to see whether the new enzyme has properties that could be competitive compared to commercially available analogues. Unfortunately, the submitted manuscript does not provide sufficient data to support this claim. It would be helpful for a wide range of readers to understand how different or superior this enzyme is to similar ones available on the market or described in the literature. At the very least, it would be interesting to compare the specific activity of bacterial alpha-amylases.

This is my main comment on the logic of the planned work.

In my opinion, the manuscript should be substantially rewritten. It should add many more details and restructure the overall logic of the presentation.

Firstly, it is clear that it would be inappropriately to discuss the substrate specificity of Alphaz before describing its biochemical properties and determining the optimal conditions for its function. Substrate specificity is typically clarified and the results demonstrated after it is known at what temperature and pH values the enzyme works best.

I propose moving subsection 2.4 to the end of the Results and Discussion section. Similarly, in the Materials and Methods section, move subsection 3.7 to the same position.

Next, I have some comments on the overall structure of the text.

  1. Abstract: The statement "Alkaline catalytic efficiency (pH 8.0) aligns with the requirements for enteric-coated drug manufacturing, while low-temperature operability (≤30 °C) preserves thermolabile natural products" sounds appealing, but it would be better if the authors described the enzyme's optimal conditions in a more traditional way.
  2. Page 2, 2nd paragraph: "To address metabolic disorders, α-amylase inhibitors are being developed as oral therapeutics to delay carbohydrate digestion and regulate postprandial blood glucose in type 2 diabetes management [23,24]." This information is redundant since the article focuses on the enzyme α-amylase itself, not its inhibitors.
  3. In the last paragraph of the Introduction, the purpose of the current study should be clearly stated. Future plans that are not part of this work should be excluded.

Results and Discussion section
4. Subsection 2.1 completely repeats subsection 3.3. It follows from the Materials and Methods section that the authors identified the strain using 16S rDNA sequence. So, where are the results of this identification? It is necessary to provide, at a minimum, data on phylogenetic analysis, as well as photos of the growth of the strain on a solid medium with the specified characteristic species features. In addition, if I understood correctly, this strain was first mentioned in [27], at the same time it was deposited in the collection of the Key Laboratory of Marine Drugs. Is this an official collection? Can anyone purchase it? The issue of strain identification should be clarified and supplemented.
5. Subsection 3.4 mentions genome-wide sequencing of the above strain and genomic annotation, but there is no data in the Results and Discussion section. I believe that this should also be made up for.
6. 2.3. Recombinant Expression and Purification of Alphaz: the A232 method for the determination of a-amylase activity is mentioned in the caption to Table 1. I'm not sure what kind of method it is.

As I mentioned before, it's also not clear what the optimal conditions are and how they were determined.

7. In Figure 2, the stripes should be numbered.

8. Table 1: How many times was protein isolation performed? What is the measurement error?

9. Page 5, last paragraph of subsection 2.4: 'In phytopharmaceutical processing, Alphaz enhances the extraction efficiency of bioactive metabolites from starch-containing medicinal plants by selectively depolymerizing structural polysaccharides [30]' - This sentence is written very strangely. It sounds as if the a-amylase studied in the current study has already been used to process starch-containing medicinal plants, although this is obviously not the case. In addition, the authors should check the correctness of the output data of the reference used.

10. The same applies to other proposed applications of Alphaz amylase. The authors only suggest that it can be used in this way, but they do not provide any evidence that they have actually tested it.

11. In the subsections (in the Results and Materials and methods) devoted to determining the optimal conditions for the functioning of the enzyme, the values of the optimal temperature vary greatly: from 35 to 50C. I suggest that the authors carefully check everything and write down in which cases each of the values was used. This should be clearly written in the Materials and methods, as well as in the Results and Discussions.

12. It is unclear why the authors believe that the optimal temperature for this enzyme is 50 °C, as it can only withstand this temperature for 5 minutes. How can such an enzyme be successfully used in practical applications?

13. Subsection 3.8 is written very carelessly. There are no subscripts in the formulas of chemical compounds, and the objectives of the work are not clearly defined. For example, instead of saying "pH optimization," it should be "determining the optimum pH values/assessing the effect of pH on enzyme function." What is "modulator studies"? Did the authors mean to assess the effect of metals and other chemical reagents on the functioning of the enzyme?

14. Why is the routine determination of enzyme activity at the very end of section 3?

Figures 4 and 5

15. Based on the description of the experiments in subsection 3.8, the panels in these figures should show residual activities, not relative ones. I think this should be fixed.

16. The description of these drawings should also be changed. The curves of alpha-amylase activity against temperature and pH are shown here, not the optima.

Conclusions

17. I believe that the conclusion about the broad thermal adaptability of the enzyme is greatly exaggerated.

Comments on the Quality of English Language

The manuscript should be carefully rewritten and English should be improved.

Author Response

Dear Reviewer,

Thank you sincerely for taking the time to review our manuscript and for providing such thoughtful and detailed feedback. Your expertise and insights have been instrumental in refining our work. Every suggestion you have made is very crucial. And we are grateful for the opportunity to improve the paper based on your suggestions.

First, we followed your suggestions to compare the kinetic parameters of our amylase with those from other sources and from polar bacterial sources, and we have also adjusted the structure of the article, placing the substrate specificity section in 2.6 and 3.8.

Response to your comments:

Comment 1: Abstract: The statement "Alkaline catalytic efficiency (pH 8.0) aligns with the requirements for enteric-coated drug manufacturing, while low-temperature operability (≤30 °C) preserves thermolabile natural products" sounds appealing, but it would be better if the authors described the enzyme's optimal conditions in a more traditional way.

Response 1: Following your suggestion, we have revised the abstract section, reducing its length and describing the optimal conditions of the enzyme in a more traditional manner.

Comment 2: Page 2, 2nd paragraph: "To address metabolic disorders, α-amylase inhibitors are being developed as oral therapeutics to delay carbohydrate digestion and regulate postprandial blood glucose in type 2 diabetes management [23,24]." This information is redundant since the article focuses on the enzyme α-amylase itself, not its inhibitors.

Response 2: Following your suggestion, we have deleted this part.

Comment 3: In the last paragraph of the Introduction, the purpose of the current study should be clearly stated. Future plans that are not part of this work should be excluded.

Response 3: Following your suggestions, we have revised and slightly modified this part of the content.

Comment 4: Subsection 2.1 completely repeats subsection 3.3. It follows from the Materials and Methods section that the authors identified the strain using 16S rDNA sequence. So, where are the results of this identification? It is necessary to provide, at a minimum, data on phylogenetic analysis, as well as photos of the growth of the strain on a solid medium with the specified characteristic species features. In addition, if I understood correctly, this strain was first mentioned in [27], at the same time it was deposited in the collection of the Key Laboratory of Marine Drugs. Is this an official collection? Can anyone purchase it? The issue of strain identification should be clarified and supplemented.

Response 4: Thank you for your suggestion. This is indeed a part that was overlooked. We have supplemented the 16S RNA sequence and growth images of the strain in the Supplementary Information (SI) and described the results of its species identification in Section 2.1 of the main text. In addition, the strain is also being studied for the biosynthesis of natural products and is an exclusive strain of the laboratory.

Comment 5: Subsection 3.4 mentions genome-wide sequencing of the above strain and genomic annotation, but there is no data in the Results and Discussion section. I believe that this should also be made up for.

Response 5: Following your suggestions, we have provided a brief description of the annotation results in Section 2.2.

Comment 6: 2.3. Recombinant Expression and Purification of Alphaz: the A232 method for the determination of a-amylase activity is mentioned in the caption to Table 1. I'm not sure what kind of method it is.

Response 6: We apologize for this elementary mistake. It should be the DNS method. We have made the correction and placed Table 1 in Section 2.7.

Comment 7: In Figure 2, the stripes should be numbered.

Response 7: Thank you very much for your suggestion. We have made the annotations.

Comment 8: Table 1: How many times was protein isolation performed? What is the measurement error?

Response 8: This indeed needs to be accurately described. We have refined the results of the three experiments. Thank you for your careful observation.

Comment 9: Page 5, last paragraph of subsection 2.4: 'In phytopharmaceutical processing, Alphaz enhances the extraction efficiency of bioactive metabolites from starch-containing medicinal plants by selectively depolymerizing structural polysaccharides [30]' - This sentence is written very strangely. It sounds as if the a-amylase studied in the current study has already been used to process starch-containing medicinal plants, although this is obviously not the case. In addition, the authors should check the correctness of the output data of the reference used.

Response 9: We apologize for the ambiguity caused by our expression. This part was merely based on assumptions from previous literature reports. Our enzyme could also be considered for use in this area in the future. This part of the content was redundant and we have removed it.

Comment 10: The same applies to other proposed applications of Alphaz amylase. The authors only suggest that it can be used in this way, but they do not provide any evidence that they have actually tested it.

Response 10: We apologize once again for this issue. We have removed it.

Comment 11: In the subsections (in the Results and Materials and methods) devoted to determining the optimal conditions for the functioning of the enzyme, the values of the optimal temperature vary greatly: from 35 to 50C. I suggest that the authors carefully check everything and write down in which cases each of the values was used. This should be clearly written in the Materials and methods, as well as in the Results and Discussions.

Response 11: We apologize for our carelessness. In the initial activity screening, we used different temperatures. Later, some parts were modified based on the previous content, which caused significant ambiguity in the temperature description. We have already made the corrections. We only used 37°C in the very beginning to check if there was any enzyme activity. Except for the thermostability test, all subsequent experiments were conducted at the optimal temperature.

Comment 12: It is unclear why the authors believe that the optimal temperature for this enzyme is 50 °C, as it can only withstand this temperature for 5 minutes. How can such an enzyme be successfully used in practical applications?

Response 12: This may be a problem with our expression. The optimal temperature of this enzyme is around 40°C. We apologize for the ambiguity caused by the unclear expression.

Comment 13: Subsection 3.8 is written very carelessly. There are no subscripts in the formulas of chemical compounds, and the objectives of the work are not clearly defined. For example, instead of saying "pH optimization," it should be "determining the optimum pH values/assessing the effect of pH on enzyme function." What is "modulator studies"? Did the authors mean to assess the effect of metals and other chemical reagents on the functioning of the enzyme?

Response 13: Thank you very much for your careful review. After reading through, we found that there were indeed problems such as disorganization and formatting errors in this part of the content. We have already made the modifications. The assessment of the effects of metal ions on enzyme activity was conducted to gain a preliminary understanding of the potential applications of this enzyme and the limitations it might face.

Comment 14: Why is the routine determination of enzyme activity at the very end of section 3?

We apologize for this elementary mistake. This part of the content has already been elaborated in Response 14: Section 3.6, so we have removed it here. Thank you very much for your careful reading.

Comment 15: Based on the description of the experiments in subsection 3.8, the panels in these figures should show residual activities, not relative ones. I think this should be fixed.

Response 15:Thank you very much for your careful observation. We indeed overlooked this part and didn't notice it before. We have changed it to residual activity.

Comment 16: The description of these drawings should also be changed. The curves of alpha-amylase activity against temperature and pH are shown here, not the optima.

Response 16: Following your suggestions, we have made some simple adjustments. Thank you.

Comment 17: I believe that the conclusion about the broad thermal adaptability of the enzyme is greatly exaggerated.

Response 17: As you pointed out, we indeed overstated the broad thermal adaptability. We have made the revisions to make the content more rigorous.

Thank you very much for taking the time to review our manuscript. Your constructive suggestions regarding the structure and logic of the article have greatly improved its rigor.

Yours

Sincerely

Reviewer 3 Report

Comments and Suggestions for Authors

This paper describes the routine cloning, purification, and characterization of a cold-adaptive amylase. However, the study lacks significant novelty. There are several weaknesses in the manuscript that need to be addressed. Therefore, I cannot recommend acceptance in its current form. The authors should carefully revise the manuscript based on my suggestions and consider resubmitting.

  1. Abstract, R&D, M&M sections: What is meant by “of animal origin”? Nocardiopsis aegyptia HDN19-252 of Antarctic animal origin. Which animal? On which body part was it found?
  2. Bioinformatics analysis is very weak. Authors should include Alphafold3 structure and highlight catalytic/binding residues.
  3. Considering the enzyme is cold-adapted there is NO analysis re its flexibility analysis or melting temperature using DSC, intrinsic fluorescence etc.
  4. Georges Feller, Rick Cavicchioli and Charles Gerday Groups have described the most important cold-adapted model alpha-amylase from Antarctic bacterium. There is no mention of any of their papers and comparison re sequence, structure and kinetic parameters with Alphaz will be very informative. Also include mammalian amylase sequence e.g. Pig in multiple sequence alignment.
  5. The most basic enzyme Michaelis-Menten kinetic parameters like kcat/Km are missing. These must be determined and compared to those from Antarctic Pseudoalteromonas haloplanktis, the model amylase.
  6. It will be very informative to determine the productivity curve of this amylase if authors are keen on showing its biotechnological potential (see Evaluating Enzymatic Productivity-The Missing Link to Enzyme Utility. Int J Mol Sci. 2022 Jun 21;23(13):6908).

Author Response

Dear Reviewer,

Thank you very much for taking the time to review our article and for your valuable and meaningful suggestions. Your expertise and insights have been instrumental in refining our work. Every suggestion you have made is very crucial. And we are grateful for the opportunity to improve the paper based on your suggestions.

Response to your comments:

Comment 1: Abstract, R&D, M&M sections: What is meant by “of animal origin”? Nocardiopsis aegyptia HDN19-252 of Antarctic animal origin. Which animal? On which body part was it found?

Respones 1: Thank you for your valuable suggestions. We did not describe this part rigorously enough before. The strain was obtained from the body of an Antarctic invertebrate, the ophiuroid, and we have added this part to Section 3.1 of the main text.

Comment 2: Bioinformatics analysis is very weak. Authors should include Alphafold3 structure and highlight catalytic/binding residues.

Respones 2:Thank you very much for the author's suggestion for improvement. We have added the structural information of the enzyme and preliminarily predicted the catalytic and binding sites. The enzyme has a typical Asp-Glu-Asp catalytic triad (Section 2.8).

Comment 3:Considering the enzyme is cold-adapted there is NO analysis re its flexibility analysis or melting temperature using DSC, intrinsic fluorescence etc.

Respones 3:Thank you for your suggestion. We have measured the Tm value of the enzyme using DSC, and the Tm value is 45.53°C.

Comment 4:Georges Feller, Rick Cavicchioli and Charles Gerday Groups have described the most important cold-adapted model alpha-amylase from Antarctic bacterium. There is no mention of any of their papers and comparison re sequence, structure and kinetic parameters with Alphaz will be very informative. Also include mammalian amylase sequence e.g. Pig in multiple sequence alignment.

Respones 4:Thank you very much for your valuable suggestions. We have compared this sequence with the amylase sequences from humans, pigs, and the Antarctic bacterial source you mentioned(Figure S2). They all possess the same catalytic triad, and we have also compared the kinetic parameters. We also made a preliminary comparison of the structure of Alphaz with that of the Antarctic cold-adapted amylase (Figure 8a). Supplemented a weak link in the article.

Comment 5:The most basic enzyme Michaelis-Menten kinetic parameters like kcat/Km are missing. These must be determined and compared to those from Antarctic Pseudoalteromonas haloplanktis, the model amylase

Respones 5:Thank you very much for your valuable suggestions. The importance of kinetic parameters is self-evident. We have supplemented the kinetic parameters of the enzyme (Table 1).

Comment 6:It will be very informative to determine the productivity curve of this amylase if authors are keen on showing its biotechnological potential (see Evaluating Enzymatic Productivity-The Missing Link to Enzyme Utility. Int J Mol Sci. 2022 Jun 21;23(13):6908).

Respones 6:Thank you very much for the suggestion regarding the productivity curve. We have also conducted a preliminary exploration under laboratory conditions (Section 2.9).

Thank you once again for taking the time to review this manuscript amidst your busy schedule.

Yours

Sincerely

Round 2

Reviewer 2 Report

Comments and Suggestions for Authors

The manuscript still needs to be improved.

  1. The result presented in subsection 2.1 is not supported by any evidence. The authors should provide a phylogenetic tree showing that the 16S RNA sequence shares the highest similarity of 99.79% with the Nocardiopsis aegyptia strain TRM86122. This is the result that should be included in the Supplementary materials, not just the RNA sequence itself. In other words, the authors are suggesting that we take their word for it that this strain is Nocardiopsis aegyptia.

  2. At the beginning of section 2.2, it is stated that the amylase sequences were compared with the genome sequences of the used strain. However, there is no evidence of the existence of this genome provided. Where can I find it and verify your claims? Are there any links to databases or publications on this subject? Also, there is no data on this topic in the work under discussion.

  3. Section 2.3 provides data on the enzyme yield without error bars. Why is there no link to Table 2 with the same data? Why has Table 2 been moved to section 2.7, where it is not mentioned in any way?

  4. In section 2.5, the numbering in panels 6b and 6c is mixed up.

  5. Why were NaCl tolerance measurements conducted at 37 and 40 °C? How is the conclusion about amylase halotolerance reached in the Conclusions section?

  6. In paragraph 2.8, it would be helpful to clarify which type of glycoside hydrolase this amylase is: retaining or reverting. Why did the authors make this determination?

  7. For the mechanism described in paragraph 2.8, it would be beneficial to provide information on the basis of which the authors determined that the mentioned residues are catalytic or substrate-binding.

  8. In general, there is little to no detailed discussion of the results in the current version of the manuscript.

  9. Paragraph 2.9 is unclear as it is presented without clear results.

  10. Section 3.2 describes the strain, which is a result, not a method or material for the Materials and Methods section. Furthermore, there are no links to Supporting Materials. How does the reader know that the authors have provided the sequence as proof and photographs of plates with the strain growing on an agarized medium.

  11. Section 3.4 explains how the researchers annotated the genome using the Rapid Annotation using Subsystem Technology (RAST) server. They also analyzed sequence homology using the BLAST algorithm from the National Center for Biotechnology Information (NCBI). To reconstruct phylogenetic relationships, they used the neighbor-joining method in MEGA 7.1.

Comments on the Quality of English Language

There are still a lot of typos and carelessness.

Author Response

Dear Reviewer,

Thank you to the reviewer for continuing to review our manuscript and for providing valuable suggestions.

Response to your comments:

Comment 1: The result presented in subsection 2.1 is not supported by any evidence. The authors should provide a phylogenetic tree showing that the 16S RNA sequence shares the highest similarity of 99.79% with the Nocardiopsis aegyptia strain TRM86122. This is the result that should be included in the Supplementary materials, not just the RNA sequence itself. In other words, the authors are suggesting that we take their word for it that this strain is Nocardiopsis aegyptia.

Respones 1: We thank the reviewer for their rigorous suggestions. We have conducted a phylogenetic analysis of the 16S RNA sequences from different genera and species in comparison with the 16S RNA sequence of HDN19-252, and the relevant results have been placed in the Supplementary Information (SI).

Comment 2: At the beginning of section 2.2, it is stated that the amylase sequences were compared with the genome sequences of the used strain. However, there is no evidence of the existence of this genome provided. Where can I find it and verify your claims? Are there any links to databases or publications on this subject? Also, there is no data on this topic in the work under discussion

Respones 2: Thank you for your suggestions. We have placed the base sequence and amino acid sequence of the amylase fragment in the Supplementary Information (SI). As for providing the complete genomic information, we will gradually release it in the future. Since this strain is unique to our laboratory and cannot be obtained from other sources, we are currently conducting many research projects centered around this strain.

Comment 3: Section 2.3 provides data on the enzyme yield without error bars. Why is there no link to Table 2 with the same data? Why has Table 2 been moved to section 2.7, where it is not mentioned in any way?

Respones 3: Firstly, we apologize for making changes without authorization and not explaining the situation in our previous response. We have now restored the table to its original position and added information on the number of purification steps and error margins.

Comment 4: In section 2.5, the numbering in panels 6b and 6c is mixed up.

Respones 4: We greatly appreciate the reviewer's meticulousness. We have made the revisions accordingly.

Comment 5: Why were NaCl tolerance measurements conducted at 37 and 40 °C? How is the conclusion about amylase halotolerance reached in the Conclusions section?

Respones 5: For enzymes of marine origin, NaCl may have certain effects on their activity and stability. When our research group works on enzymes of marine origin, we usually measure this aspect as well.

The choice of 37 and 40℃ is based on thermal stability. The enzyme shows better stability at 37 ℃, while its stability significantly decreases at 40 ℃. Therefore, we selected these two distinct temperatures to check whether NaCl would have a certain impact on the enzyme's stability.

Regarding the conclusion on salt tolerance, we measured the enzyme activity changes under different NaCl concentrations and found that the enzyme still maintained good activity at a concentration of 1.5 M. Thus, we concluded that the enzyme has a certain degree of salt tolerance.

Comment 6: In paragraph 2.8, it would be helpful to clarify which type of glycoside hydrolase this amylase is: retaining or reverting. Why did the authors make this determination?

Respones 6: This amylase belongs to the classic glycoside hydrolase family 13 and has the ability to hydrolyze α-1,4-glycosidic bonds. We also conducted sequence alignment with the most classic Antarctic-derived cold-adapted α-amylase and other sources of amylase. They all possess the same catalytic residues, namely the Asp-Glu-Asp catalytic triad. Moreover, we performed a preliminary structural comparison with the classic Antarctic-derived α-amylase, and the two share a high degree of similarity in overall structure.

We would like to retain this part of the content, as other reviewers have expressed a desire to see some structural information in the article. We apologize for not informing you about the additional changes requested by the other reviewers in our previous response.

Comment 7: For the mechanism described in paragraph 2.8, it would be beneficial to provide information on the basis of which the authors determined that the mentioned residues are catalytic or substrate-binding.

Respones 7: We apologize for not elaborating on the determination of the catalytic sites. The catalytic sites were identified through sequence alignment with the classic Antarctic-derived cold-adapted α-amylase and other sources of amylase. The binding sites were also obtained through alignment and speculation. The results of the sequence alignment have been placed in the Supplementary Information (SI).

Comment 8: In general, there is little to no detailed discussion of the results in the current version of the manuscript.

Respones 8: Thank you for the reviewer's suggestions. This is indeed a part that was lacking in our paper. We have now discussed the potential application fields based on the actual characteristics of the enzyme and proposed preliminary improvement measures. The relevant content has been placed in Section 2.10 of the article.

Comment 9: Paragraph 2.9 is unclear as it is presented without clear results.

Respones 9: We apologize for the poor description of the enzyme productivity curve. This section was added based on the suggestions of other reviewers. We have re-analyzed this part of the content in conjunction with the curve results.

Comment 10: Section 3.2 describes the strain, which is a result, not a method or material for the Materials and Methods section. Furthermore, there are no links to Supporting Materials. How does the reader know that the authors have provided the sequence as proof and photographs of plates with the strain growing on an agarized medium.

Respones 10: The ophiuroid samples were collected at 61°42′28′′ S, 57°38′22′′ W in Antarctica.. The broken animal samples were cultured on ISP2 solid medium by the dilution spread plate method. After 7 days of cultivation, single colonies were picked and streaked in three zones to obtain pure single colonies.
The above content has been updated in Section 3.2.

Comment 11: Section 3.4 explains how the researchers annotated the genome using the Rapid Annotation using Subsystem Technology (RAST) server. They also analyzed sequence homology using the BLAST algorithm from the National Center for Biotechnology Information (NCBI). To reconstruct phylogenetic relationships, they used the neighbor-joining method in MEGA 7.1.

Respones 11:  The researchers annotated the genome using the Rapid Annotation using Subsystem Technology (RAST) server. To validate annotations and explore sequence homology, we employed NCBI BLAST tools with stringent parameters (high query coverage) against curated databases, identifying conserved domains and potential horizontal gene transfer events. Phylogenetic relationships were reconstructed using the neighbor-joining method in MEGA 7.1, involving multiple sequence alignment of conserved genes , model selection, to assess tree robustness, ultimately linking functional annotations to evolutionary patterns across species.If there are any aspects where our understanding is inadequate, please feel free to contact us. We will make further revisions according to your suggestions.

We would like to thank you once again for taking the time to review our manuscript. Your suggestions have been immensely helpful in shaping the direction and overall quality of our article.

Yours

Sincerely

Reviewer 3 Report

Comments and Suggestions for Authors

Authors have carried out new experiments which is commendable. There are still many issues with the manuscript mainly to do with the presentation and referencing.

There is nothing wrong with the productivity curve. It shows the productivity over time under the conditions used. Rewrite the discussion in view of the productivity paper I suggested. 

Something doesn't seem right about kcat, Km values of human and pig. Alphaz values should also be compared with cold-adapted amylase from model Pseudomonas haloplanktis. Generally cold-adapted enzymes have a higher kcat than mesophilic homologues. Why human amylase have a higher kcat? Also, human and pig kcat values should be similar.

Authors may find to put all kinetic values (kcat, Km, Topt, Tm etc) under one table and compare them with model cold-adapted amylases and one mesophilic homologues. 

Section 3.9 heading is all wrong. It should not be "Substrates Specificity Analysis of Alphaz " but "Bioinformatic analysis". 

Logically Table 1 should come after Table 2.

The legends to all figures should have more details about the method used and conditions.

Comments on the Quality of English Language

English expression needs to be improved a lot before the manuscript can be published.

Author Response

Dear Reviewer,

Thank you very much for your timely review and evaluation. It allows me to quickly receive your feedback and begin the revision. I am also very grateful for this opportunity to improve the quality of the manuscript. Incorporating the suggestions from the other reviewer, we have now placed the original Table 2 in Section 2.3 and added a new discussion section 2.10. We hope you understand these changes.

Response to your comments:

Comment 1: There is nothing wrong with the productivity curve. It shows the productivity over time under the conditions used. Rewrite the discussion in view of the productivity paper I suggested. 

Respones 1: Regarding the productivity curve, the trend has been described, the reasons have been explained, and a speculative conclusion has been drawn. The relevant content is in Section 2.9 and has been highlighted in red. Thank you for your valuable suggestions.

Comment 2: Something doesn't seem right about kcat, Km values of human and pig. Alphaz values should also be compared with cold-adapted amylase from model Pseudomonas haloplanktis. Generally cold-adapted enzymes have a higher kcat than mesophilic homologues. Why human amylase have a higher kcat? Also, human and pig kcat values should be similar.

Authors may find to put all kinetic values (kcat, Km, Topt, Tm etc) under one table and compare them with model cold-adapted amylases and one mesophilic homologues. 

Respones 2: In response to your comments on the kinetic parameters, we have followed your suggestions and compared our α-amylase with the cold-adapted α-amylase from Antarctic sources and the mesophilic α-amylase from porcine sources by placing their kcat, Km, Tm, Topt and kcat/Km in the same table. We have also reviewed the literature and found that thekcat values of α-amylases from human and porcine sources are similar.

Comment 3: Section 3.9 heading is all wrong. It should not be "Substrates Specificity Analysis of Alphaz " but "Bioinformatic analysis".  

Respones 3: We apologize for such a basic mistake. We have made the corrections. Thank you for your careful review.

Comment 4: Section 3.9 heading is all wrong. It should not be "Substrates Specificity Analysis of Alphaz " but "Bioinformatic analysis".  

Respones 4: We apologize for such a basic mistake. We have made the corrections. Thank you for your careful review.

Comment 5: Logically Table 1 should come after Table 2.  

Respones 5: Thank you for your suggestions on the logic of the table. We have made the revisions.

Comment 6: The legends to all figures should have more details about the method used and conditions..  

Respones 6: Thank you for your suggestions. We have added the experimental conditions and detection methods in the figure legends.

We are very grateful to the reviewer for taking the precious time to review the manuscript and providing valuable and timely comments, which have given us the opportunity to improve the quality of the manuscript.

Yours

Sincerely

Round 3

Reviewer 2 Report

Comments and Suggestions for Authors

The current version of the manuscript seems to be suitable for publication. However, there are some minor errors that need to be corrected.

  1. The text should contain references to all the figures and tables provided.

  2. In the captions of the figures where phylogenetic trees are shown, it is necessary to include information about bootstrap values, the scale bar representing the number of nucleotide substitutions per site, and the names of the studied microorganism and the outgroup species.

  3. If you mention in the Materials and Methods section that you have annotated the genome, you should also provide the results. If you do not want to share them, be sure to clearly state where you obtained the sequence. The general rule for writing scientific articles is that all claims MUST be supported by data!

Comments on the Quality of English Language

Please check carefully all the text.

Author Response

Dear Reviewer,

First of all, please allow me to express my gratitude to you. This is my first time writing this type of article, and there were numerous errors. It was you who repeatedly pointed them out and offered suggestions, enabling me to improve the article step by step. I have indeed gained some insights into the logic of writing through this process, for which I am truly thankful.

Response to your comments:

Comment 1: The text should contain references to all the figures and tables provided.

Respones 1: Thank you very much for your suggestions. All the figures and tables have now been referenced in the text. The newly added references that were not previously marked have been highlighted in red in the main text.

Comment 2:In the captions of the figures where phylogenetic trees are shown, it is necessary to include information about bootstrap values, the scale bar representing the number of nucleotide substitutions per site, and the names of the studied microorganism and the outgroup species.

Respones 2: In accordance with your suggestions, the caption for Figure S1 has been added, and the phylogenetic tree for strain identification has been reconstructed. The phylogenetic tree in Figure 1 in the main text has also been added with a scale bar representing the number of amino acid substitutions. Due to the size of the figure, it has been placed in the Supplementary Information as Figure S4. Thank you very much for your suggestion.

Comment 3: If you mention in the Materials and Methods section that you have annotated the genome, you should also provide the results. If you do not want to share them, be sure to clearly state where you obtained the sequence. The general rule for writing scientific articles is that all claims MUST be supported by data!

Respones 3: We thank the reviewer for the suggestion. We have applied for the GenBank number for the nucleotide sequence of the enzyme and placed it in Section 2.3 in red. The genome sequence and nucleotide sequence have also been included in the Supplementary Information (SI). We greatly appreciate the reviewer's understanding.

We are very grateful to the reviewer for taking the precious time to review the manuscript and providing valuable and timely comments, which have given us the opportunity to improve the quality of the manuscript.

Yours

Sincerely

Fuhao Liu

Reviewer 3 Report

Comments and Suggestions for Authors

Only some of my suggestions have been taken into account. The most important comment I emphasized was referencing, which the authors totally ignored. I am pasting it again here:   In the revised version, referencing is not done properly. For example, none of the methods like Alphafold3, productivity etc, have been referenced. This is not appropriate. Every technique described under M&M Section MUST have a reference so that the readers can replicate the experiments. Also, key references on model cold-adapted alpha-amylase from Feller Groups #31 and 32 are cited in a strange way in the text. All of them are clumped together [19-32]. This is not the way to do referencing. These key references from Feller, Cavicchioli Groups should be discussed upfront in the Introduction section. Carefully recheck all references in the list and text.

Secondly, I said the captions to all figures must be checked, revised and details must be added to them. For example, Figure 4 is a Differential Scanning Calorimetry (DSC) thermogram and no mention is made of this method in M&M section. This is carelessness. The full method should be given in M&M section along with the reference (Protein Engineering, Design & Selection vol. 23 no. 10 pp. 769–780, 2010, doi:10.1093/protein/gzq051). Check carefully for other omissions. 

In Table 2, Topt (40 deg) is less than the Tm (45.5) which indicates that in this enzyme, the active site unfolds first, followed by the unfolding of the entire structure. For better discussion, consult the above reference.

Comments on the Quality of English Language

Can be improved. For example, what "broken animal" means? Which animal? You described the name in the rebuttal but did not add here.

Author Response

Dear reviewer

Here, I would like to express our gratitude on behalf of all the authors. This is my first time writing this type of article, and there were many gaps and logical flaws. It was your timely feedback and suggestions that enabled me to improve the article step by step. Through this process, I gradually gained knowledge of enzymology and learned some experimental techniques. For all of this, I am truly thankful. Moreover, I would like to explain that some issues might not have been addressed well in the revision due to my insufficient understanding.

Response to your comments:

Comment 1: In the revised version, referencing is not done properly. For example, none of the methods like Alphafold3, productivity etc, have been referenced. This is not appropriate. Every technique described under M&M Section MUST have a reference so that the readers can replicate the experiments. Also, key references on model cold-adapted alpha-amylase from Feller Groups #31 and 32 are cited in a strange way in the text. All of them are clumped together [19-32]. This is not the way to do referencing. These key references from Feller, Cavicchioli Groups should be discussed upfront in the Introduction section. Carefully recheck all references in the list and text.

Respones 1: In accordance with your suggestions, the cold-adapted amylase from Antarctic sources has been mentioned and briefly discussed in the Introduction. All methods have been properly referenced in the Materials and Methods section.

Comment 2: Secondly, I said the captions to all figures must be checked, revised and details must be added to them. For example, Figure 4 is a Differential Scanning Calorimetry (DSC) thermogram and no mention is made of this method in M&M section. This is carelessness. The full method should be given in M&M section along with the reference (Protein Engineering, Design & Selection vol. 23 no. 10 pp. 769–780, 2010, doi:10.1093/protein/gzq051). Check carefully for other omissions.

Respones 2: Thank you for your reminder. We have checked the figure captions and added information about the environmental conditions of some operations. The more detailed procedures and methods have been placed in the Materials and Methods section. Additionally, the content regarding the DSC measurement has also been added.

Comment 3: In Table 2, Topt (40 deg) is less than the Tm (45.5) which indicates that in this enzyme, the active site unfolds first, followed by the unfolding of the entire structure. For better discussion, consult the above reference.

 Respones 3: Thank you for raising such professional questions. We have revised the content in accordance with the references, and the revised content is as follows: We found that for both Alphaz and AMY_PSEHA, their optimal temperatures are lower than the Tm. The potential reasons for this may be that the active site region has higher flexibility, causing the active site to unfold and expose first. Another possible reason is the reduction of salt bridges and hydrophobic interactions. For example, the Antarctic enzyme AMY_PSEHA has fewer salt bridges and hydrophobic clusters than the porcine enzyme, resulting in a lack of rigidity around the active site and its preferential unfolding[25]. The preferential unfolding of the active site of the Antarctic α-amylase is a structural trade-off for its cold adaptability, rooted in the decoupling of high flexibility of the catalytic center from overall stability. This mechanism provides a key target for the rational design of psychrophilic enzymes[31].

Thank you once again for your suggestions and for giving us the opportunity to revise the article.

Yours

Sincerely

Fuhao Liu

Round 4

Reviewer 3 Report

Comments and Suggestions for Authors

The revised manuscript is much better than the original version. Figure 4 legend still needs to be revised to include that the graph was generated using differential scanning calorimeter (DSC) as I suggested before.

Comments on the Quality of English Language

can be improved during editorial editing.

Author Response

Dear reviewer

Thank you very much for your timely review and suggestions. Your suggestions are always timely, giving me ample time to make revisions according to your advice. Thank you very much.

Response to your comments:

Comment 1:The revised manuscript is much better than the original version. Figure 4 legend still needs to be revised to include that the graph was generated using differential scanning calorimeter (DSC) as I suggested before.

Respones 1: In accordance with your suggestions, we have revised the legend for Figure 4 to xxx.In accordance with your suggestions, we have revised the legend for Figure 4 to The melting temperature (Tm) of Alphaz. Heated from 25°C to 100°C within 75 minutes. The graph was generated using differential scanning calorimetry (DSC).

Thank you once again for your suggestions and for giving us the opportunity to revise the article.

Yours

Sincerely

Fuhao Liu
